# Species-wide whole genome sequencing reveals historical global spread and recent local persistence in *Shigella flexneri*

Thomas R Connor[1,2]*[†], Clare R Barker[1†], Kate S Baker[2†], François-Xavier Weill[3], Kaisar Ali Talukder[4], Anthony M Smith[5], Stephen Baker[6,7,8], Malika Gouali[3], Duy Pham Thanh[6,7,8], Ishrat Jahan Azmi[4], Wanderley Dias da Silveira[9], Torsten Semmler[10,11], Lothar H Wieler[10,11], Claire Jenkins[12], Alejandro Cravioto[13], Shah M Faruque[4], Julian Parkhill[2], Dong Wook Kim[14], Karen H Keddy[5], Nicholas R Thomson[2,8]

[1]Cardiff School of Biosciences, Cardiff, United Kingdom; [2]Pathogen Genomics, Wellcome Trust Sanger Centre, Cambridge, United Kingdom; [3]Unité des Bactéries Pathogènes Entériques, Institut Pasteur, Paris, France; [4]Centre for Food and Water Borne Diseases, International Centre for Diarrhoeal Disease Research, Dhaka, Bangladesh; [5]Centre for Enteric Diseases, National Institute for Communicable Diseases and Faculty of Health Sciences, University of the Witwatersrand, Johannesburg, South Africa; [6]The Hospital for Tropical Diseases, Wellcome Trust Major Overseas Programme, Oxford University Clinical Research Unit, Ho Chi Minh City, Vietnam; [7]Centre for Tropical Medicine, Nuffield Department of Clinical Medicine, Oxford University, Oxford, United Kingdom; [8]The London School of Hygiene and Tropical Medicine, London, United Kingdom; [9]Department of Genetics, Evolution, and Bioagents, Institute of Biology, University of Campinas, São Paulo, Brazil; [10]Centre for Infection Medicine, Institute of Microbiology and Epizootics, Freie University, Berlin, Germany; [11]Robert Koch Institute, Berlin, Germany; [12]Gastrointestinal Bacteria Reference Unit, Public Health England, London, United Kingdom; [13]Global Evaluative Sciences, Inc., Seattle, United States; [14]Department of Pharmacy, School of Pharmacy, Hanyang University, Ansan, Republic of Korea

*For correspondence: ConnorTR@cardiff.ac.uk

†These authors contributed equally to this work

**Competing interests:** The authors declare that no competing interests exist.

**Abstract** *Shigella flexneri* is the most common cause of bacterial dysentery in low-income countries. Despite this, *S. flexneri* remains largely unexplored from a genomic standpoint and is still described using a vocabulary based on serotyping reactions developed over half-a-century ago. Here we combine whole genome sequencing with geographical and temporal data to examine the natural history of the species. Our analysis subdivides *S. flexneri* into seven phylogenetic groups (PGs); each containing two-or-more serotypes and characterised by distinct virulence gene complement and geographic range. Within the *S. flexneri* PGs we identify geographically restricted sub-lineages that appear to have persistently colonised regions for many decades to over 100 years. Although we found abundant evidence of antimicrobial resistance (AMR) determinant acquisition, our dataset shows no evidence of subsequent intercontinental spread of antimicrobial resistant strains. The pattern of colonisation and AMR gene acquisition suggest that *S. flexneri* has a distinct life-cycle involving local persistence.

## Introduction

Once a major cause of mortality and morbidity in Europe and the US prior to the widespread provision of reliable sanitation systems and clean drinking water, bacterial dysentery caused by *Shigella* spp.,

**eLife digest** Dysentery is a disease in which the intestine becomes inflamed due to infection by bacteria, viruses or other microbes. Of the bacteria that can cause dysentery, bacteria called *Shigella* are most often responsible. Humans can acquire *Shigella* through contaminated food or water. Over the last century, improvements to sanitation combined with access to clean drinking water and better food hygiene have decreased the number of cases of dysentery in many countries. However, the disease continues to be common in low-income countries, especially in young children.

One species of *Shigella* bacteria, called *S. flexneri*, causes far more cases of dysentry than other species of *Shigella*. Across the world, there are many different strains of *S. flexneri*, but it is not clear how these strains are related to each other, or how variable the genes that they carry are—known as genetic diversity.

Here, Connor, Barker, Baker et al. used a technique called whole genome sequencing to map the evolutionary relationships of over 300 *S. flexneri* samples collected from around the globe over the past 100 years. This revealed that the bacterial strains can be split into seven groups that each have distinct geographic ranges and combinations of genes that enable the bacteria to infect humans. Many of the strains of bacteria within these groups seem to have colonized, and remained in, quite small geographic areas over long periods of time. This is different to other *Shigella* species, which appear to have spread between continents far more frequently over much shorter timescales.

Connor, Barker, Baker et al.'s findings reveal that *S. flexneri* is more diverse than other *Shigella* bacteria, and suggest that the ability of strains to persist in local areas may have contributed to the species' long-term success. These results point towards the importance of the provision of clean water in the fight against *S. flexneri*, and underline the need for a greater understanding of how disease-causing bacteria colonize and interact with the local environment.

remains a significant infection in low-income countries (*Kotloff et al., 2013*). While the Shigellae are, phylogenetically, *Escherichia coli*, they were originally classified as separate species based upon shared disease and biochemical phenotypes that marked them out as distinct from other *E. coli* strains—a distinction that is still reflected in their species nomenclature because of continued global medical importance. The 'genus' *Shigella* consists of four species (*Shigella flexneri*, *Shigella sonnei*, *Shigella boydii* and *Shigella dysenteriae*) causes approximately 165 million new infections globally per year (*Kotloff et al., 1999*; *Ram et al., 2008*), which have previously been estimated to result in up to 1 million deaths annually (*Kotloff et al., 1999*, *2013*). The vast majority of cases and fatalities occur in low to middle-income countries in children under the age of 5 years (*Kotloff et al., 1999*; *von Seidlein et al., 2006*). The preponderance of these cases are attributable to endemic disease caused by the species *S. flexneri* (*Vinh et al., 2009*; *Ud-Din et al., 2013*; *Livio et al., 2014*). Despite the importance of *S. flexneri* as an etiological agent of diarrheal disease globally, little is known about its detailed population structure. This knowledge gap is a substantial limitation as the ability to accurately track bacterial pathogens is a cornerstone of effective surveillance and downstream public health interventions.

This poor understanding of *S. flexneri* is partly a result of a lack of high-resolution tools for subtyping this species. Traditionally *S. flexneri* strains are subdivided based on the antigenic variation of the O-antigen component of the bacterial lipopolysaccharide (LPS) using typing antisera and the slide agglutination method (serotyping). Serotyping currently subdivides isolates into serotypes or subserotypes by the use of type-specific and group factors antisera (*Edwards and Ewing, 1986*; *Sun et al., 2011*). Although serotyping still forms the central vocabulary for describing this species it is now widely agreed that many genes that determine serotype are encoded on horizontally transmissible genetic elements (*Allison and Verma, 2000*), thus facilitating serotype switching. Therefore the extent to which core characteristics of the species that are currently inferred based on serotype relate to the phylogenetic relationships of strains remains open to question.

Through limited genetic and genomic studies we know that *Shigella* spp. represent distinct clades that fall within the *E. coli* species complex (*Pupo et al., 2000*; *Yang et al., 2005*, *2007*; *Ashton et al., 2014*; *Sahl et al., 2015*). These studies show the existence of two distinct *S. flexneri* lineages, with one lineage including only a single serotype (*S. flexneri* 6) (*Choi et al., 2007*) that clusters within species

*S. boydii* (*Yang et al., 2007*) and possesses a different LPS O-antigen. The second monophyletic *S. flexneri* group contains representatives of all other *S. flexneri* serotypes (1–5, X, Y) and is responsible for the majority of *S. flexneri* disease; it is this main lineage that is examined here.

Whole genome sequencing has been used successfully to uncover key aspects of the provenance as well as global and regional epidemiology of other *Shigella* species. Work on the now rarely isolated (*von Seidlein et al., 2006*), *S. dysenteriae* type 1 has shown it to be an epidemic pathogen characterised by sporadic, large scale outbreaks that spread rapidly and widely via a series of intercontinental transmissions, often associated with war or famine (*Rohmer et al., 2014*). Studies examining the dominant endemic *Shigella* species of industrialised and newly industrialised countries, *S. sonnei*, have also revealed a species that is highly clonal (*Karaolis et al., 1994*; *Holt et al., 2012*). *S. sonnei* evolved in Europe ~300 years ago and has recently moved out of this region via intercontinental spread as a single, rapidly evolving lineage, establishing new, local, populations in countries as they industrialise (*Holt et al., 2012*, *2013*). The contrast between the distribution of the two endemic *Shigella* species, *S. sonnei* and *S. flexneri*, has posed the question why is it that two different species predominate under very different socio-economic conditions (*Kotloff et al., 1999*)? *S. flexneri* remains the dominant species in low-income countries (multiple studies report over 50% of all cases of shigellosis [*Kotloff et al., 1999*, *2013*; *Livio et al., 2014*]), whereas *S. sonnei* is the most commonly isolated *Shigella* species in industrialised countries (77% of cases as reported by *Kotloff et al., 1999*) where there is better sanitation and access to clean food and drinking water. The reasons for these differences are poorly understood, as are specific transmission patterns that may account for the variation in the dominant *Shigella* species in these differing settings.

To further our understanding of the evolution, population structure and phylogeography of *S. flexneri* we gathered a representative global collection of 351 isolates of *S. flexneri*, spanning serotypes 1–5, X, Xv and Y, collected from the contemporary principal foci of endemic disease; Africa, Asia and South and Central America as well as historical isolates from reference collections dating back to 1914 (*Supplementary file 1*). We performed whole genome sequencing on the sample set to provide a basis for exploring the relationship of strain, serotype and geography. Our analysis reveals that *S. flexneri* is composed of phylogenetically distinct lineages, with each lineage holding similar levels of diversity to the entire *S. sonnei* species. We observe that the natural history of *S. flexneri* is characterised by long term (in some cases over 100 years) colonisation of individual countries demonstrating that it is far older, and far more diverse than *S. sonnei*. From this, our analysis uncovers further key differences between the population structure of these two species; providing new clues as to the reasons for the persistence, and most recently decline of *S. flexneri*.

## Results

### The genomic population structure of *S. flexneri*

To determine a detailed phylogeny of this species we mapped the sequence reads from the 351 *S. flexneri* strains to the concatenated reference genome of *S. flexneri* strain 301, including its virulence plasmid (VP), to detect single nucleotide polymorphisms (SNPs). SNPs falling in transposases, IS elements, repeats or regions identified as being recombinant (*Figure 1—figure supplement 1*) and unlikely to reflect the underlying phylogeny of the bacterium, were excluded from this analysis. We identified 67,981 SNPs among the 352 genomes (including reference genome), with 63,186 of these on the chromosome and 4795 on the VP. Following the removal of recombinant regions (almost entirely concentrated in phage, IS elements and *S. flexneri* Pathogenic Islands, *Figure 1—figure supplement 1*), 55,662 SNPs remained in total, with 53,078 on the chromosome and 2584 on the VP. From these SNPs we inferred a maximum likelihood phylogeny showing that the population of *S. flexneri* is composed of several pylogenetically distinct lineages (*Figure 1*, *Figure 1—figure supplement 2*). To unambiguously subdivide the species based on shared patterns of sequence variation, we used the software package Bayesian analysis of population structure (BAPS) (*Cheng et al., 2011*, *2013*) to identify robust phylogenetic groups (PGs) within *S. flexneri*. This subdivided the population into seven PGs, concordant with the phylogeny (*Figure 1*). Performing the BAPS analysis on alignments of the chromosome and VP collectively or individually resulted in the same pattern of clustering, reflecting that the VP phylogeny precisely mirrors that of the chromosome, indicating co-evolution (*Figure 1—figure supplement 3*).

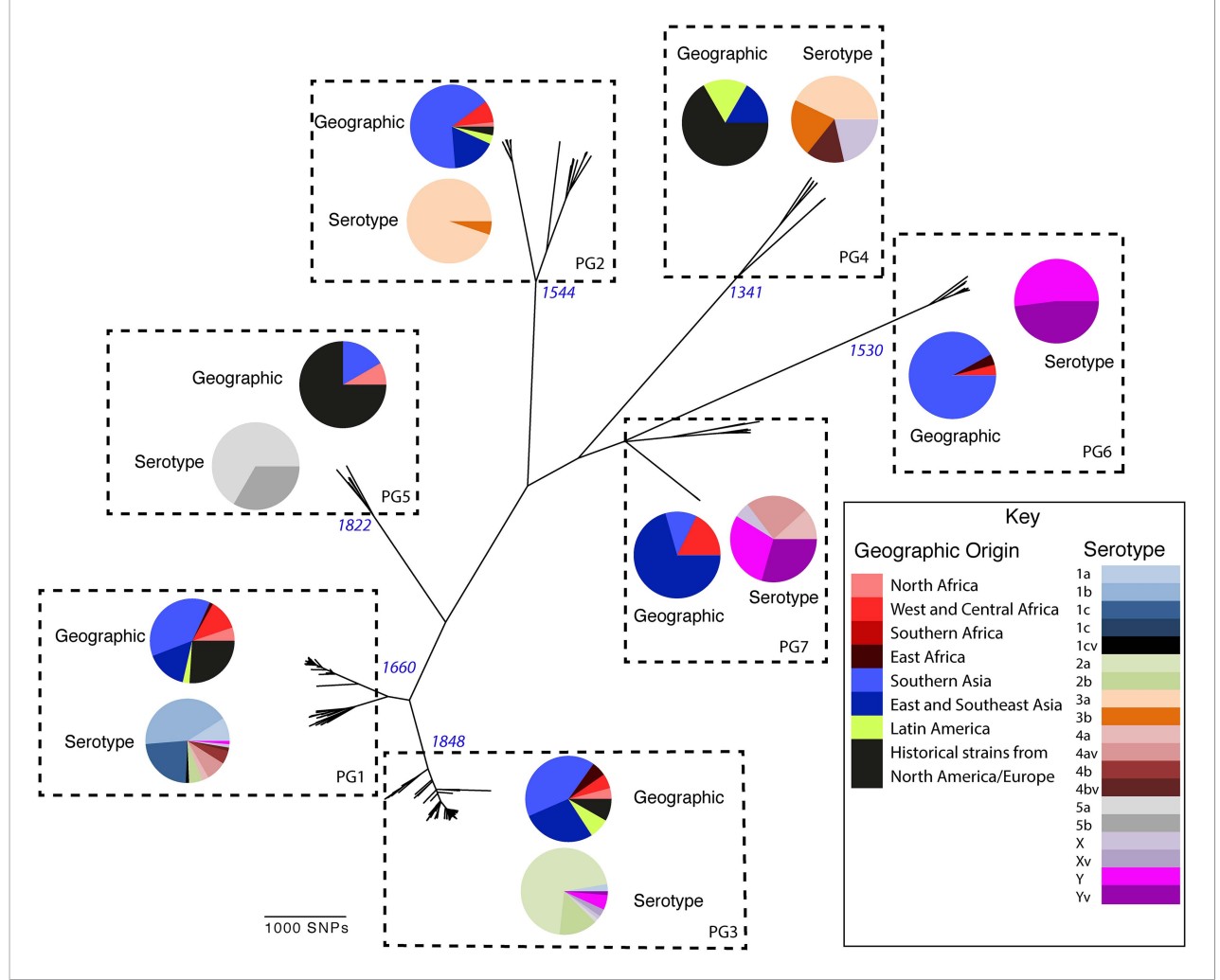

**Figure 1**. Maximum likelihood phylogeny for *Shigella flexneri* isolates including serotypes 1–5, X and Y produced from the results of mapping sequence reads against the genome of *S. flexneri* 2a strain 301, with recombination removed. Phylogenetic groups (PGs) determined by Bayesian analysis of population structure clustering are boxed within dotted lines, with the geographic and serotype composition of isolates in each PG being inlaid as pie charts.

The following figure supplements are available for figure 1:

**Figure supplement 1**. Location of segments detected as recombinant.

**Figure supplement 2**. *S. flexneri* species tree, with the number of single nucleotide polymorphisms (SNPs) per branch.

**Figure supplement 3**. Co-evolutionary relationships of the *S. flexneri* genome and virulence plasmid (VP).

**Figure supplement 4**. Maximum Clade Credibility trees generated using Bayesian evolutionary analysis by sampling trees (BEAST) for PG 1.

**Figure supplement 5**. Maximum Clade Credibility trees generated using BEAST for PG 2.

**Figure supplement 6**. Maximum Clade Credibility trees generated using BEAST for PG 3.

**Figure supplement 7**. Maximum Clade Credibility trees generated using BEAST for PG 5.

## The natural history of *S. flexneri*

Genetic clustering of the population and the phylogenetic tree revealed a species that is composed of seven discrete PGs, separated by considerable evolutionary distance (ranging from 1138 to 8430 SNPs between PGs—*Figure 1—figure supplement 2*), this analysis provided only limited information regarding the natural history of the organism. Most of the PGs contained organisms from several geographic regions, and significantly all PGs contained samples collected over a period of at least 60 years. To combine the geographic and temporal metadata with the genomic information, we used Bayesian evolutionary analysis by sampling trees (BEAST) (*Drummond and Rambaut, 2007*) to reconstruct the temporal and geographical history of the major PGs. Inferring evolutionary rates and estimating the dates of the most recent common ancestor (MRCA) of individual PGs, we found that rates of mutation were similar among PGs (between $6.46 \times 10^{-7}$ substitutions per site per year [PG1] to $9.54 \times 10^{-7}$ substitutions per site per year [PG2]). These mutation rates are consistent with previous estimates for *S. flexneri* ($3.2 \times 10^{-7}$ substitutions per site per year [*Zhang et al., 2014*]), *S. sonnei* ($6.0 \times 10^{-7}$ substitutions per site per year [*Holt et al., 2012*]) and *S. dysenteriae* ($6.52 \times 10^{-7}$ substitutions per site per year [*Rohmer et al., 2014*]). Using these data we predicted the MRCA of each PG, finding that the median age of the MRCA for all PGs was between 150 and 900 years ago (*Figure 1*, *Figure 1—figure supplements 4–7*). Of the PGs, PG1, 2, 4 and 6 are the oldest lineages with median BEAST estimate of MRCAs of these groups dating to between 1341 and 1659. Contrastingly, PG5 and PG3, the latter containing the majority of serotype 2a isolates—one of the key suggested *S. flexneri* vaccine targets—are much younger, and these PGs have median estimated MRCAs dating to 1822 (PG5) and 1848 (PG3); contemporaneous with the MRCA of the emergence of *S. sonnei* from Europe. However, unlike *S. sonnei*, the emergence of new *S. flexneri* PGs did not appear to result in displacement or replacement of isolates from other PGs; rather the older PGs have persisted and continue to cause disease alongside the newer PGs, as evidenced by the fact that every PG contains at least one strain collected since 2008 (*Supplementary file 1*).

## The intercontinental spread of *S. flexneri*

One of the most striking features of the *S. sonnei* population was the recent intercontinental spread of the organism. Analysing our dataset using the discrete states phylogeographic analysis implemented in BEAST we observed a contrasting natural history of *S. flexneri* PGs 1–7 with respect to *S. sonnei* (*Holt et al., 2012*). We observed considerable genomic diversity in *S. flexneri* isolates collected from the same region. Regions where *S. flexneri* is endemic simultaneously supported at least two populations of *S. flexneri* originating from distinct PGs; with the Indian subcontinent having geographically-monophyletic sublineages from four different PGs (*Figure 1*, *Figure 1—figure supplements 4–6*, *Figure 2*). None of these sublineages are recently introduced into these regions; most geographically monophyletic sublineages within the PGs had MRCAs in the range of 25–150 years. As well as observing isolates from different genomic backgrounds coexisting in the same geographic area, our data also show that sublineages from the same PG can also persist contemporaneously in the same geographical region, over an extended period of time (*Figure 1—figure supplements 4, 6*).

Despite this structure of long-term phylogeographic association on a population wide level, we also observe that some PGs were highly geographically restricted and appear to have spread to only a limited number of regions. For example, PG2 has a MRCA dated to 1550 (95% CI: 1340–1725), with Asian isolates appearing in both sublineages of this cluster (*Figure 1—figure supplement 5*). The MRCA for the Asian isolates was estimated to be 1544; however the African isolates form a single cluster that has a more recent MRCA—dated to 1885 (95% CI: 1833–1923); a result that implies that PG2 *S. flexneri* was likely introduced into Africa from Asia at least 130 years ago. However, it is also important to note that while there is always a possibility of sample bias; within our dataset there is a clear signal that the African and Asian lineages are distinct, and have been for some time. Other samples may change this view, however, the sampling strategy we used was to collect samples globally—capturing a cross section of current disease with a historical view of the population of *S. flexneri*—without a specific focus on a single serotype grouping or geographic region. The fact that clear clusters within the population associated with geographic origin do appear would not be expected by chance.

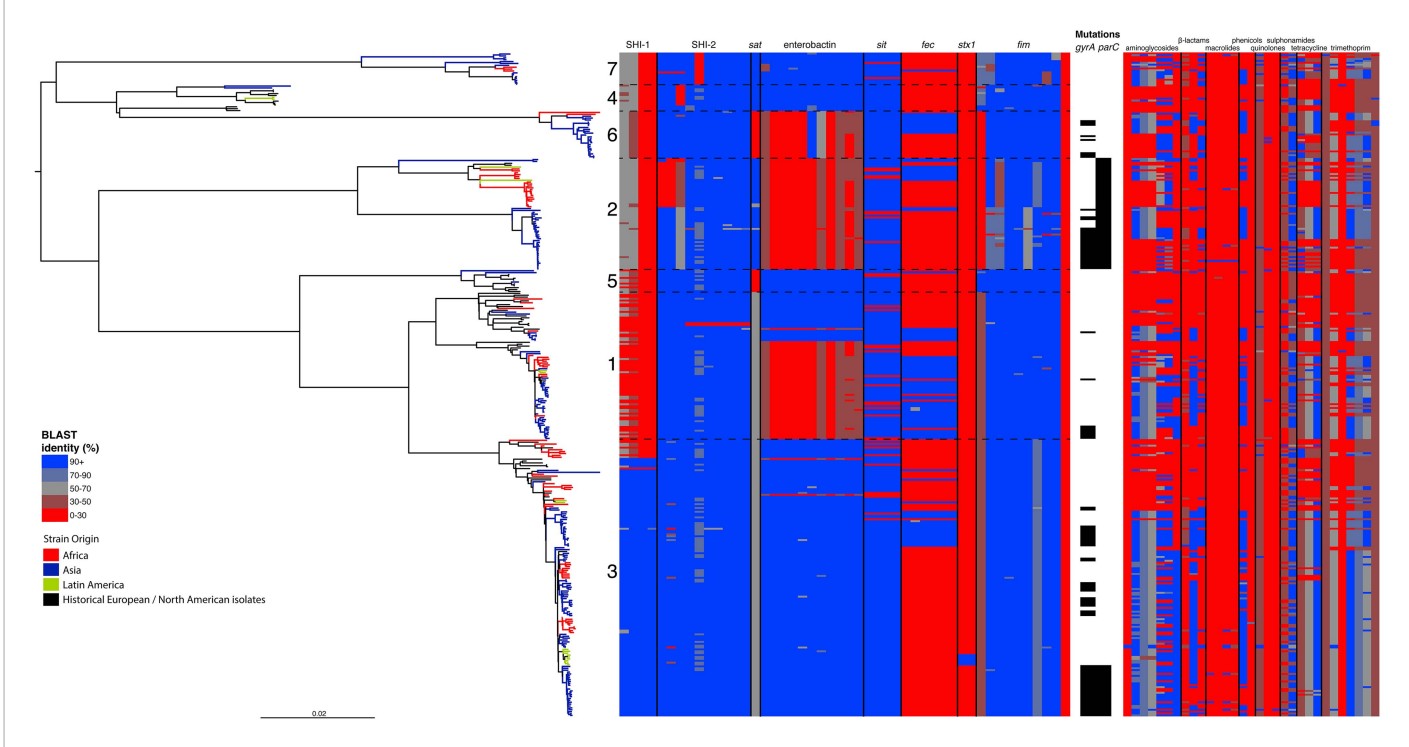

**Figure 2**. Correlation of isolate phylogeny with pathogenicity and antimicrobial resistance (AMR) determinants. The midpoint-rooted maximum likelihood phylogenetic tree shows PGs, with tips and terminal branches collared by continent of origin. Tracks adjacent to each isolate show the percentage BLAST identity of the best hit in the sample assembly against key virulence and AMR determinants. Isolates with mutations in the *gyr/par* genes have black bars in the relevant tracks. The virulence determinants shown are SHI-1 (*pic*, *sigA*, *set1AB*), SHI-2 (*shiABCDE*, *iucABCD*, *iutA*), *sat*, enterobactin (*entABECFD*, *fepABCDG*), *sitABCD*, *fecEDCBAR*, *stx1ab*, *fimZBCHGFDEAY*, and the AMR genes are *aac(3)-II*, *aadA1*, *aadA2*, *aadA5*, *strA*, *strB*, *sat*1 (aminioglycosides), $bla_{CTX-M-24}$, $bla_{OXA-1}$, $bla_{TEM-1}$, (β-lactams) *ermB*, *msrE*, *mphA*, *mphE*, (macrolides) *catA1*, *catB1*, (phenicols) *qacEΔ1*, *qnrS1*, (quinolones) *qepA*, *sul1*, *sul2* (sulphonamides) *tetA(A)*, *tetA(D)*, *tetA(B)* (tetracyclin), *dfrA17*, *dfrA3b*, *dfrA1*, *dfrA5*, *dfrA14* and *dfrA8* (trimethoprim).

The following figure supplements are available for figure 2:

**Figure supplement 1**. Results of molecular serotyping, displaying the distribution of MLST, molecular serotype, and the distribution of defining genes (according to key, top left) among isolates.

**Figure supplement 2**. Maximum likelihood phylogeny of an alignment of the concatenated nucleotide sequences the enterobactin locus of 13 genes (34,732 NT; containing *entABCDEFS*, *fepABCDG* and *fes*).

**Figure supplement 3**. Correlation of isolate phylogeny with AMR determinants, showing only the SRL-MDRE-associated loci *aadA1*, $bla_{OXA-1}$, *cat* and *tetA(B)*.

## The genome of *S. flexneri* is stable, with recombination mostly limited to mobile elements

Perhaps surprisingly for a gastrointestinal pathogen of the *E. coli* complex, at least on a population level, we found very limited evidence of large-scale species-wide recombination within the genome (*Figure 1—figure supplement 1*), with *S. flexneri* displaying a recombination profile more in line with the lifestyle-restricted ST131 than with members of other gastrointestinal pathovars such as ETEC (*McNally et al., 2013*). Within the virulence repertoire, where recombination could be identified, it was generally present among all isolates within a single PG(s), indicating that the recombination was likely to be ancestral occurring prior to PG divergence. Furthermore, most evident recombination is restricted to previously identified mobile genetic elements (*Figure 1—figure supplement 1*).

By examining known chromosomally encoded virulence determinants, it is clear that many known virulence functions are ubiquitous to all PGs (*Figure 2*). The distribution of virulence determinants

indicated that many of the genes/islands whose loss is known to attenuate *S. flexneri* virulence (such as the VP [*Sansonetti et al., 1982*] and aerobactin [*Lawlor et al., 1987*; *Wyckoff et al., 2009*]) were acquired early in its evolutionary history and have been subsequently retained. However, some variation in the composition of virulence determinants is also evident. For example, Sat, a Serine Protease Autotransporter, is found in PGs 2, 4, and 7, but absent from PGs 1, 3, 5 or 6 (*Figure 2*). Although the latter PGs possess a genetic scar suggesting that Sat was present ancestrally and has been subsequently lost.

We also find that the SHI-1 pathogenicity island—a distinguishing feature of serotype 2a (the dominant serotype causing disease world-wide [*Kotloff et al., 1999*])—was acquired once (between 1848 and 1882) near to the root of PG3. SHI-1 carries genes encoding three key virulence associated proteins—Pic, (*Henderson et al., 1999*) SigA (*Al-Hasani et al., 2000*), and the *S. flexneri* enterotoxin ShET1 (*Fasano et al., 1997*). Pic, which is also found in Enteroaggregative *E. coli*, encodes a secreted protease that has been shown to have mucinase activity and has been suggested to be involved in haemagglutination and serum resistance (*Henderson et al., 1999*). SigA is thought to be involved in fluid accumulation (*Al-Hasani et al., 2000*), while the *S. flexneri* enterotoxin ShET1 has also been shown to be involved in fluid accumulation in rabbit ileal loop models (*Fasano et al., 1997*), and has been proposed as being the causative agent of the voluminous watery diarrhoea characteristic of a *S. flexneri* serotype 2a infection (*Noriega et al., 1995*; *Fasano et al., 1997*). Although SHI-1 is only present in PG3 (135 of 146 PG3 isolates included carried this island), it is present in the range of serotypes found within this phylogroup—including multiple isolates of serotypes 1a, 2a, 2b and 5a. This demonstrates that the toxin is not peculiar to strains carrying a particular serotype, but rather to strains within a PG that are predominantly serotype 2a; consistent with reports of a small number of non-2a serotypes reported to encode this toxin (*Noriega et al., 1995*). Our genomic data further confirm that these cases were not simply mis-serotyped, but likely represent cases of sporadic serotype switching amongst isolates within PG3.

More generally, we observe evidence for recombination amongst the genes responsible for serotype (see *gtr Figure 1—figure supplement 1*), which remains the de-facto method for describing strains of *Shigella*. When cross comparing the classical and in silico molecular serotyping schemes (*Sun et al., 2011*) we saw no significant difference in the serotypes predicted (assessed using a Kolmogorov–Smirnov test). However, our results do demonstrate that while certain serotypes make up the majority of certain clusters (*Figure 1*, *Figure 2—figure supplement 1*), all of the PGs contained multiple serotypes. This indicates that although the core genome is remarkably stable, serotype switching does occur undermining the principle that serotype can be used either as a basis for describing the properties of a lineage or for epidemiological surveillance and tracking. These results also imply that vaccines targeting a particular backbone using a serotype (such as a 2a vaccine to target SHI-1 encoding isolates from PG3) could be susceptible to the effects of serotype switching.

In addition to our observations around SHI-1 and the VP we also observed some variation in virulence factor complement within PGs principally for Iron uptake functions; the *sit* (*Runyen-Janecky et al., 2003*) and *fec* loci (*Luck et al., 2001*) and enterobactin biosynthesis operon. Across our *S. flexneri* tree it is apparent that these have contrasting patterns of gain and loss; *sit* is ancestral, but was sporadically lost (unexpectedly, as it has been reported to be in all clinical isolates previously [*Runyen-Janecky et al., 2003*]), while *fec* has been gained sporadically by isolates within PG3, as well as being gained (and retained) on separate occasions by isolates in PG1, 2 and 6. Of additional interest is the enterobactin operon; which encodes a high affinity iron siderophore (*O'Brien and Gibson, 1970*). It has previously been reported that in *S. flexneri* the enterobactin genes are 'rarely utilized' (*Schmitt and Payne, 1988*; *Wyckoff et al., 2009*; *Reuter et al., 2014*); however, our data show that the enterobactin genes are present across PGs 3, 5, 7 and 4, as well as a subset of related isolates within of PG1. Examining the enterobactin gene specifically, we observed the same phylogenetic topology as that observed for the whole genome tree (*Figure 2*, *Figure 2—figure supplement 2*); suggesting this element is ancestral, and has been lost by a subset of PGs. The ubiquitous nature of these enterobactin genes suggests that these genes are more significant than 'rarely utilized'; as they have been retained by multiple PGs, over a long timescale.

In addition to these PG-wide patterns, within PG3 we observed six tightly clustered isolates that also carry a Shiga toxin 1a (Stx1a)-encoding phage (*Figure 2*) which is identical to the recently reported φPOC-J13 (accession number KJ603229) (*Gray et al., 2014*). These isolates were collected between 2003 and 2008 from Latin America (Haiti, French Guiana and the Dominican Republic). This is

consistent with these findings charting the emergence of Stx1a-producing *S. flexneri* in the region (*Gray et al., 2014*). Given that the reference phage clusters phylogenetically within our sequenced isolates it is likely that the phage was acquired once in this sublineage of PG3. The dating analysis on the Stx-1-containing PG3 sublineage showed that the phage could have been acquired no later than 1998 (*Figure 1—figure supplement 6*) suggesting that strains carrying this phage may have been present but undetected in the population for 5–10 years before the earliest reported isolation of Stx1a-producing *S. flexneri*.

## Antimicrobial resistance determinants have been gained and retained locally on multiple occasions independently by multiple lineages

Antimicrobial resistance (AMR) has been shown to be a strong influence on the recent evolutionary history of many bacterial pathogens (*Mutreja et al., 2011*; *Holt et al., 2012*; *Okoro et al., 2012*; *He et al., 2013*; *Mather et al., 2013*). Screening for the presence of known AMR genes (*Figure 2*) we found that AMR gene distribution was highly variable both within and across PGs (*Figure 2*). Unlike *S. sonnei*, we observe little evidence that the acquisition of AMR-related loci is linked to the establishment or replacement of dominant lineages in any geographic location. While published evidence does exist of one major, on-going, clonal outbreak of *S. flexneri* in high income countries that has been shown to be the result of a combination of specific epidemiological, AMR and behavioural factors within the MSM community (*Baker et al., 2015*), examining our dataset focused around historical isolates and those with an origin in low-income countries, we observe limited historical evidence of global expansion as a result of AMR. In contrast we observe the widespread acquisition of AMR determinants independently across the tree. The most widespread AMR loci are found on the multidrug resistance element (MDRE), part of the Shigella resistance locus-pathogenicity island (SRL-PAI) with variants found in representatives from all PGs. Our analysis suggests that the SRL-PAI has been introduced independently on at least nine occasions (*Figure 2—figure supplement 3*). In six of these it has been maintained in subsequent lineages for protracted periods of time. There is, however, no evidence of isolates carrying SRL-PAI spreading outside of their geographic region, and, once acquired, in a number of cases it appears that the SRL-MDRE may have been subsequently lost (*Figure 2—figure supplement 3*). Beyond the SRL-PAI we observe 23 internal branches where non SRL-PAI resistance determinants have been acquired (*Figure 2*), depicting a distinctive wave-like pattern of resistance determinant acquisition over time (*Figure 3*). The earliest resistance genes observed in the population code for sulphonamide resistance (1950s), followed by streptomycin and tetracycline (1960s) resistance, with β lactamases arising first in strains isolated in the 1970s (*Figure 3A*). Notably, we observe that in all cases within *S. flexneri* the strains acquiring AMR determinants remain geographically restricted, although on a population level we also observe an upward trend in AMR determinant possession over time, in all PGs (*Figure 3B*). We also observe that whilst AMR is widespread in the 21st century, we can isolate *S. flexneri* strains that do not harbour large numbers of AMR determinants; this is in marked contrast to observations within the *S. sonnei* population.

## Discussion

Our results present the clearest and most complete overview of the temporal and geographic patterns that underpin the population structure of the species *S. flexneri* obtained to date. It provides a genetic framework and evolutionary context for studies looking within single countries or looking at individual outbreaks or lineages of interest. These data contrast with that of the other *Shigella* species for which we have global population data, *S. sonnei* and *S. dysenteriae*; with the pattern of long term colonisation, diversity and coexistence in *S. flexneri* lineages appearing to share more similarities with pathogenic *E. coli* variants such as ETEC (*von Mentzer et al., 2014*) than other *Shigella* species. It is evident from the presented data that, at a population level, *S. flexneri* is characterised by long-term colonisation of endemic regions, with limited evidence of intercontinental transmission within the last 30 years. Our data also suggest that endemic countries support a diverse population of *S. flexneri*, and have done for some time, implying long-term local-transmission/colonisation in those settings. This is in direct contrast to *S. sonnei*, where historically the population is characterised by recent colonisation of countries and dominant pandemic lineages, with frequent inter-country and inter-continental transmission and repeated strain replacement (*Holt et al., 2012*).

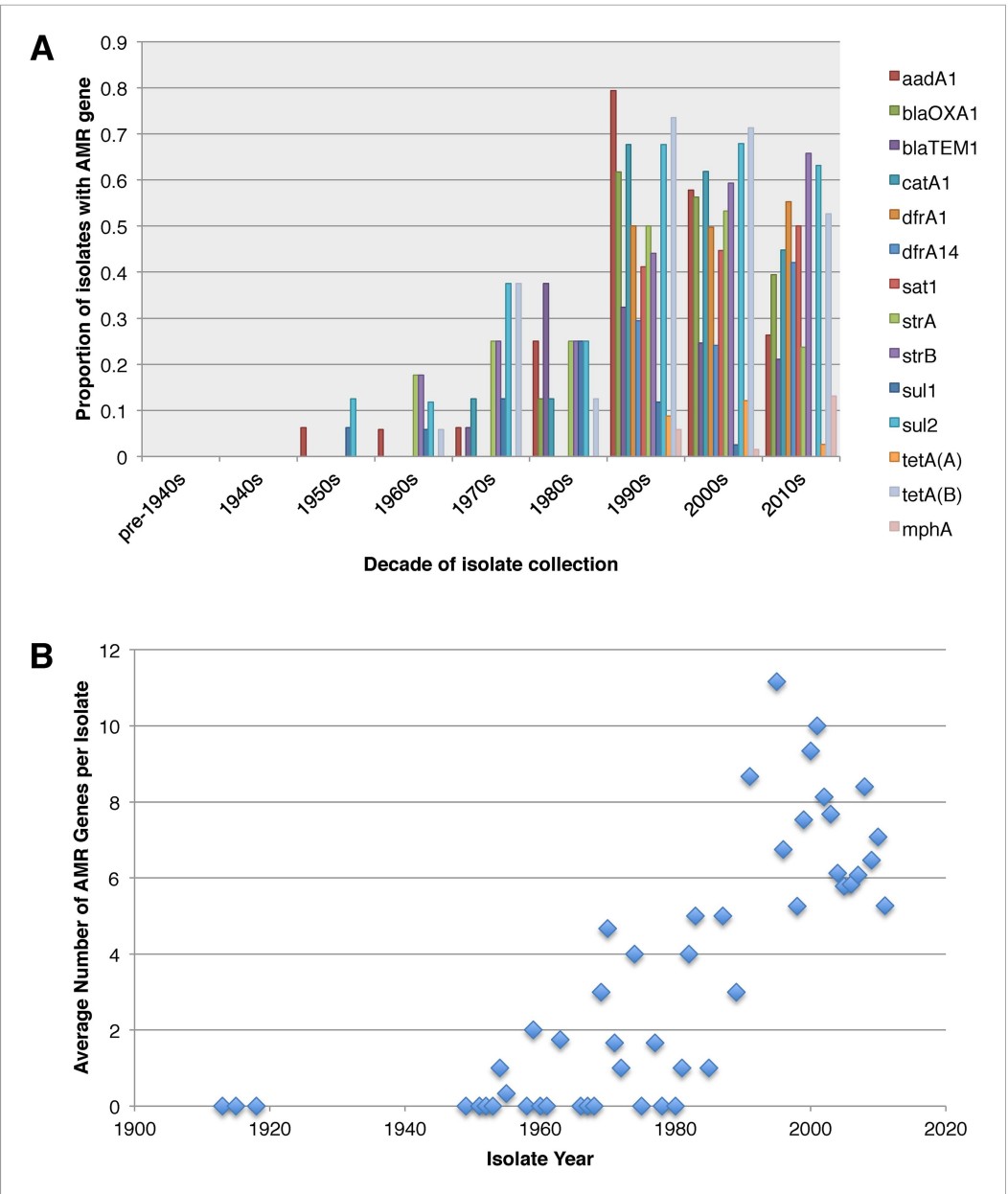

**Figure 3**. Graphs showing the pattern of AMR presence within our dataset. (**A**) Graph showing the proportion of isolates from each decade that contain the AMR genes. (**B**) Graph showing the average number of resistance genes found in each isolate collected, by year.

It is also evident that while AMR has had significant in the natural history of both species, the effects of AMR acquisition on the population are contrasting. For *S. sonnei* the acquisition of multiple AMR genes was strongly associated with lineage replacement by dominant resistant lineages, on a global scale. Whereas, for *S. flexneri*, we observe a population where there have been over 30 independent acquisitions of AMR determinants at a local level; but these acquisitions generally result in limited global spread (*Figure 2*). This implies that while AMR may be a significant factor in maintaining existing lineages within specific locales we do not see widespread evidence of AMR acquisition leading to the pandemic displacement of other pre-existing lineages. Moreover this observation also extends to the simultaneous acquisitions of multiple AMR genes *en bloc*, such as those associated with the SRL-PAI. The acquisition of this locus appeared to have occurred in lineages

that, within this dataset, show no subsequent evidence of travel outside of their original geographic area. It may be that the transient nature of AMR gene carriage is the result of rapid gain and loss during periods of differential selection, consistent with *S. flexneri* persisting in the environment where it is conceivable that selection by AMR could be less significant. Despite this lack of evidence of rapid global spread of AMR, on a population level we did observe both an on-going trend of increasing resistance in *S. flexneri*, and evidence that this pathogen acquires resistance determinants rapidly, often keeping the AMR determinants for protracted periods. However, we also continue to observe recently collected strains with a low number of resistance determinants—suggesting that strains with a limited resistance repertoire are continuing to persist within the wider population. This observation suggests that although there is evidently some selective advantage to carrying AMR determinants, as evidenced by the increasing level of resistance across the pathogen population, it is not an absolute requirement for modern *S. flexneri*.

The contrast between the observed population structure, natural history and role of AMR in the narrative of *S. flexneri* and *S. sonnei* may be related to the lifecycle of these two organisms. It has previously been shown that *S. flexneri* strains can be recovered from the environment in endemic countries (*Faruque et al., 2002*) unlinked to any outbreak, and that under experimental conditions *S. flexneri* can survive for several months in water at room temperature (*Uyanik et al., 2008*), simulated environmental water (*Hendricks and Morrison, 1967*; *Islam et al., 1996*) and upon foodstuffs (*Islam et al., 1993*). Furthermore, there are multiple examples of *S. flexneri* outbreaks where the outbreak strain was traced back to, and isolated from local water sources (*Swaddiwudhipong et al., 1995*; *He et al., 2009*; *Saha et al., 2009*). In addition, at least one study has been undertaken that has demonstrated that within Bangladesh *S. flexneri* disease risk was persistent throughout the year and associated with environmental factors (geographic location/provision of flood defences) (*Emch et al., 2008*). When these studies are combined with both our results and additional observations that the provision of clean water and good sanitation correlates with a reduction in the rates of *S. flexneri* disease (*Esrey et al., 1985*), we believe that the implication is that in an environment where water is frequently persistently contaminated with faeces, *S. flexneri*, in effect, persists in the environment, and has been doing so over the long term, in countries around the world. A mechanism of environmental persistence would explain our observations around the longevity of lineages of *S. flexneri* in endemic countries and provide an obvious transmission mechanism. When considering the temporal and phylogeographic analyses presented here in this context, we think that the evidence supports the concept, suggested previously, that in endemic countries the lifecycle of *S. flexneri* disease is analogous to that of *Vibrio cholerae* (*Faruque et al., 2002*); with part of the pathogens infectious cycle being in contaminated water, from which it periodically emerges to cause disease. While our data suggests that an environmental stage may be important for *S. flexneri*—whose reduced incidence has previously been correlated with the provision of clean water and improved sanitation (*Esrey et al., 1985*), the inverse is true for *S. sonnei*. It has been suggested that the O-antigen carried by *Plesiomonas shigelloides* (another organism isolated from faecally contaminated water)—that is immunologically indistinguishable from the *S. sonnei* O-antigen may help to explain the contrasting global distribution of *S. sonnei* and *S. flexneri* infections (*Sack et al., 1994*). Exposure to *P. shigelloides*, through drinking contaminated ground water has been hypothesized to stimulate low level cross protective immunity against *S. sonnei*, explaining why it is rarely found in the resource poor settings but its incidence increases with water sanitation. If the *P. shigelloides* hypothesis is proved to be true, it may be that contaminated water holds the key to both the increase in *S. sonnei* and the decrease in *S. flexneri* observed as countries develop.

Collectively, our analyses outline a historic pathogen with a stable core genome that comes equipped with a repertoire of virulence determinants that have enabled it to colonise, and persist, in multiple locations for hundreds of years. However, this life-cycle is at odds with current patterns of human development; with temporal analysis revealing limited evidence of intercontinental transmission and local colonisation within the last 30 years. It may be that this change is reflective of the fact that over the course of the 20th Century, many countries have industrialised; with increasing provision of clean water and sanitation possibly disrupting the traditional transmission route of *S. flexneri*. However, despite its decline in industrialised countries, where transmission of this pathogen by faecally contaminated food and water is well controlled, *S. flexneri* continues, as it has done for hundreds of years, to be a source of disease infecting millions in the low-income nations of the world.

# Materials and methods

## Samples

Contemporaneous strains isolated from patients were collected from accredited diagnostic or Public Health laboratories located in Bangladesh (ICDDRB, n = 114), South Africa (National Institute for Communicable Diseases, Johannesburg, South Africa, n = 29), and, for Francophone Africa and Latin America, from the French National Reference Centre for *E. coli*, *Shigella*, and *Salmonella* (Institut Pasteur, n = 54) and the Hospital for Tropical Diseases, Ho Chi Minh City, Vietnam (n = 34). A historical selection of isolates was obtained from the Institut Pasteur (n = 41) and The International Vaccine Institute and Hanyang University, Korea (n = 62), to provide a stronger temporal signal for the BEAST analysis. Additionally, we also included the HPA type strains (n = 16) (*Ashton et al., 2014*) and NCTC1 (*Baker et al., 2014*). Serotyping was performed on isolates using the standard tests used locally by the submitting lab (using either commercial or locally produced, polyclonal and monoclonal typing antisera), and the classical serotypes obtained were checked, and in most cases confirmed, using a molecular serotyping schema (*Sun et al., 2011*).

## Sequencing

DNA was extracted by collecting laboratories and sequenced at the Wellcome Trust Sanger Institute to a minimum of 50-fold coverage using an Illumina HiSeq 2000 with multiplexing of 96 samples per flow cell using 100 bp paired end reads. Sequencing data was submitted to the ENA (Genome accession numbers provided in *Supplementary file 1*). To perform subsequent analyses around the AMR and virulence gene determinants, sequence data was assembled de novo using Velvet (*Zerbino and Birney, 2008*), with assemblies improved using iCORN (*Otto et al., 2010*) and the Velvet Columbus module.

## Bioinformatics analysis

Phylogenetic and population genetic relationships were determined based on a similar approach to that employed by *Holt et al. (2012)* for *S. sonnei*. Sequence (FASTQ) files for each isolate in the study were mapped back against the reference *S. flexneri* serotype 2a strain 301 concatenated with its complete VP (Accession numbers: AE005674, AF386526). A whole genome alignment was produced by SNP-calling isolates, and regions that were recombinant were identified using the software package Gubbins (*Croucher et al., 2014*). The recombinant regions were subsequently manually checked and their content identified. Following removal of recombinant regions, mapping coverage was checked manually and variable sites were extracted. The resulting SNP alignment was used to infer a phylogeny using RAxML version 7.4 with a General Time Reversible model and gamma correction (n = 4) for among site rate variation (*Stamatakis, 2006*). The alignment produced prior to the removal of recombinant regions (66,524 bp in length) was passed to BAPS to identify phylogenetic clusters (*Corander et al., 2004*, *2008*).

The individual plasmid alignment was generated by extracting the part of the combined plasmid/chromosome alignment that contained plasmid sequence only. This alignment was then analysed on its own. Following SNP calling, this produced a SNP alignment of 4784 bp that was used to generate an unrooted phylogeny and perform BAPS clustering (as above) within the plasmid population.

To examine the temporal and phylogeographic distribution of the samples we made use of BEAST, providing taxa subsets of the recombination-free alignment and providing geographic information and sample dates. The parameters were as follows: a Bayesian skyline model for population growth was used, with a log-normally distributed clock rate. For each PG BEAST was run across five chains of 100,000,000 generations each, sampled every 1000 generations. Convergence was determined by visual inspection of MCMC parameters across the chains. All parameter ESS values were ≥200. The parameter and tree estimates were combined using the LogCombiner and TreeAnnotator components of the BEAST package, with the first 10% of states in each chain discarded as burn-in, and then produced a Maximum Clade Credibility tree from the combined file, which was visualized with FigTree.

The presence of virulence and AMR determinants (*Figure 2*) was determined by BLAST against reference loci as described previously (*Reuter et al., 2014*). For the AMR genes (*Figure 2*) the

presence of mutations in the DNA gyrase and topoisomerase IV protein sequences were assessed by inspection of de novo assembled sequences.

## Acknowledgements

TRC, KSB, JP and NRT were all funded by the Wellcome Trust, grant number 098051. CRB was funded by a Wellcome Trust/NISCHR ISSF project at Cardiff University. The Bioinformatics analysis performed at Cardiff used resources funded by the MRC (Grant ref: MR/L015080/1) and Cardiff University (storage funded by the Cardiff University Research Infrastructure Fund). FXW and MG are funded by the Institut Pasteur, the Institut de Veille Sanitaire, and by the French Government 'Investissement d'Avenir' program (Integrative Biology of Emerging Infectious Diseases Laboratory of Excellence, grant no. ANR-10-LABX-62-IBEID). DWK was supported by the grant 2012R1A2A2A01009741 from National Research Foundation (NRF) of Korea. We thank David Harris and the sequencing teams at the Sanger Institute for sequencing the samples. We thank Dr Munirul Alam, icddr,b for the helpful discussions and we thank Isabelle Carle, Monique Lejay-Collin, Corinne Ruckly from the Institut Pasteur for their excellent technical assistance.

## Additional information

### Funding

| Funder | Grant reference | Author |
|---|---|---|
| Wellcome Trust | 098051 | Thomas R Connor, Kate S Baker, Julian Parkhill, Nicholas R Thomson |
| Medical Research Council (MRC) | MR/L015080/1 | Thomas R Connor |
| Cardiff University | | Thomas R Connor, Clare R Barker |
| Institut Pasteur | | François-Xavier Weill, Malika Gouali |
| Institut de veille sanitaire | | François-Xavier Weill, Malika Gouali |
| Government of the French Republic | ANR-10-LABX-62-IBEID | François-Xavier Weill, Malika Gouali |
| National Research Foundation of Korea | 2012R1A2A2A01009741 | Dong Wook Kim |
| Wellcome Trust/National Institute for Social Care and Health Research ISSF grant to Cardiff University | | Clare R Barker |

The funders had no role in study design, data collection and interpretation, or the decision to submit the work for publication.

### Author contributions

TRC, DWK, Conception and design, Acquisition of data, Analysis and interpretation of data, Drafting or revising the article; CRB, KSB, Analysis and interpretation of data, Drafting or revising the article; F-XW, KAT, Conception and design, Acquisition of data, Drafting or revising the article, Contributed unpublished essential data or reagents; AMS, SB, SMF, Acquisition of data, Drafting or revising the article, Contributed unpublished essential data or reagents; MG, DPT, IJA, Acquisition of data, Drafting or revising the article, Contributed unpublished essential data or reagents; WDS, Drafting or revising the article, Contributed unpublished essential data or reagents; TS, LHW, Analysis and interpretation of data, Drafting or revising the article, Contributed unpublished essential data or reagents; CJ, Conception and design, Drafting or revising the article; AC, Conception and design, Acquisition of data, Drafting or revising the article; JP, Conception and design, Drafting or revising the article, Contributed unpublished essential data or reagents; KHK, NRT, Conception and design, Analysis and interpretation of data, Drafting or revising the article, Contributed unpublished essential data or reagents

### Author ORCIDs

Thomas R Connor, http://orcid.org/0000-0003-2394-6504

## Additional files

### Supplementary file

• Supplementary file 1. Table containing strain information, accession numbers for the strains used in this study along with the blast identities for the virulence and AMR genes displayed in *Figure 2*.

### Major datasets

The following datasets were generated:

| Author(s) | Year | Dataset title | Dataset ID and/or URL | Database, license, and accessibility information |
|---|---|---|---|---|
| The *Shigella* Genome Sequencing Consortium | 2015 | Global Diversity of Shigella Species | http://www.ncbi.nlm.nih. gov/bioproject/? term=PRJEB2846 | Publicly available at the NCBI BioProject (Accession no: PRJEB2846). |
| The *Shigella* Genome Sequencing Consortium | 2015 | Shigella sonnei and flexneri from around the world | http://www.ncbi.nlm.nih. gov/bioproject/204320 | Publicly available at the NCBI BioProject (Accession no: PRJEB2460). |
| The *Shigella* Genome Sequencing Consortium | 2015 | Shigella flexneri from around the world | http://www.ncbi.nlm.nih. gov/bioproject/? term=PRJEB2542 | Publicly available at the NCBI BioProject (Accession no: PRJEB2542). |

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
