## [Decision Letter]

Thank you for submitting your work entitled “Species-wide whole genome sequencing reveals historical global spread and recent local persistence in *Shigella flexneri*” for peer review at *eLife*. Your submission has been favorably evaluated by Diethard Tautz (Senior editor) and two reviewers, one of whom, Allan McNally, has agreed to share his identity.

The reviewers have discussed the reviews with one another, and the editor has drafted this decision to help you prepare a revised submission.

Summary:

The authors present a comprehensive analysis of the evolution of *Shigella flexneri*. The work is excellently conducted and the conclusions drawn in the manuscript are sound and solidly supported by the data. The contrast presented in the manuscript between *S. flexneri* and *S. sonnei* make this more than just another species phylogeny paper, showing very interesting insights into the effect of socio-demographic influences on how pathogenic lineages of bacteria evolve.

Essential revisions:

1) Looking at the section on AMR and the phylogeny it seems that PG3, which contains the serotype 2a clinically dominant strains, is the most recently emerged of the phylogroups. When one looks at the mapping of the AMR genes the low-diversity bottom half of PG3 appears to have a near identical AMR pattern. So whilst you state there is no effect of AMR on the evolution of *S. flexneri* lineages and dissemination, one could argue that PG3 shows signs of clonal expansion of an AMR lineage? Please clarify.

Further to this point on the discussion that AMR is not an absolute requirement for modern *S. flexneri*. Is it not plausible that the patchwork appearance of AMR seen in the majority of the PGS may be due to transient carriage of AMR genes? Such that they are only maintained for short periods of time when selection for them is great, but then they are rapidly lost during the hypothesised environmental phase of their life cycle?

2) We consider the suggestion of a different life style or a different life-cycle between *S. flexneri* and other *Shigella* species as a very interesting hypothesis and very important to the field, if true. However, such an interpretation of the genomics data ideally requires further experimental support. As far as we know, *Shigella* has no environmental reservoir and human is its only host. You cite a paper reporting the isolation of *S. flexneri* from the environment (16); however, similar numbers of *S. dysenteriae* were also isolated from these environmental samples and therefore, do not support a different life style of *S. flexneri* than other *Shigella* species. Furthermore, different virulence genes (ipaBCD) from the virulence plasmid were missing in these so-called environmental isolates, but were present in clinical isolates, suggesting avirulent phenotype of these isolates.

It would therefore be prudent to provide more evidence for an environmental persistence of *S. flexneri*, for example from genome sequences of food or environmental isolates. Are these available in your collection? Further, it might be useful to check for clues in the genome why *S. flexneri* might be more persistent in the environment. Are there any differences in carbon or nitrogen utilization pathways (different number of pseudogenes) in relation to *S. Sonnei?*

Alternatively, we suggest a test in the lab. A controlled comparison of the survival of *Shigella* species in water can provide strong experimental support for such an assumption and possibly explain the different epidemiology of *S. flexneri* vs. *S. Sonnei*.

In case that neither the bioinformatic analysis provide additional clues nor the experimental approach is feasible, we would ask you to clearly explain the caveats associated with your assumption of different life styles.

3) A slight weakness is that recent *S. flexneri* isolates from European origin are underrepresented in the collection. It appears that the most recent European strain (E2610/72) was from 1972. As global tourism from endemic countries to Europe increased greatly in the last 40 years or so, facilitating a possible intercontinental transmission might be overlooked if recent isolates are not included. For example of a recent outbreak in Europe, please see: M L Borg Ongoing outbreak of *Shigella flexneri* serotype 3a in men who have sex with men in England and Wales, data from 2009-2011. Eurosurveillance, Volume 17, 2012. Accordingly, we should like to ask you to assess the possibility to add more recent isolates from Europe in your study.

---

## [Author Response]

*1) Looking at the section on AMR and the phylogeny it seems that PG3, which contains the serotype 2a clinically dominant strains, is the most recently emerged of the phylogroups. When one looks at the mapping of the AMR genes the low-diversity bottom half of PG3 appears to have a near identical AMR pattern. So whilst you state there is no effect of AMR on the evolution of* S. flexneri *lineages and dissemination, one could argue that PG3 shows signs of clonal expansion of an AMR lineage? Please clarify*.

We agree that this needs clarifying, and would say that we did not intend to suggest that there is ‘no effect’ as an absolute, rather within our dataset we do not believe there is strong historical evidence for the effect of AMR on the species as a whole. In the specific case highlighted the SNP distance between most isolates in PG3 is >100 SNPs. This is not consistent with a very recent clonal expansion but as you point out it does not exclude the possibility of this happening. Furthermore we do know of one unique case where – in the MSM population, as indicated below – an *S. flexneri* clone has spread by clonal expansion related to AMR. To address this point we have modified a section of our AMR results section to read:

“While published evidence does exist of one major, on-going, clonal outbreak of *S. flexneri* in high income countries that has been shown to be the result of a combination of specific epidemiological, antimicrobial resistance and behavioral factors within the MSM community [5], examining our dataset focused around historical isolates and those with an origin in low-income countries, we observe limited historical evidence of global expansion as a result of antimicrobial resistance. In contrast we observe the widespread acquisition of AMR determinants independently across the tree.”

*Further to this point on the discussion that AMR is not an absolute requirement for modern* S. flexneri*. Is it not plausible that the patchwork appearance of AMR seen in the majority of the PGS may be due to transient carriage of AMR genes? Such that they are only maintained for short periods of time when selection for them is great, but then they are rapidly lost during the hypothesised environmental phase of their life cycle?*

It is certainly possible. Our intention in stating that AMR is not an absolute requirement was a general commentary on the fact that over the whole population the pattern of AMR gain and loss was markedly different from other pathogenic *Shigella* species, such as *S. sonnei*, where strain/lineage replacement appears to now be driven by the acquisition of AMR. This pattern is clearly not present within *S. flexneri* .

The reviewer points to a possible mechanism that would drive this observation. Although this is complicated by the fact that a subset of the AMR genes are carried by the (chromosomal) SRL-PAI which is more stable, we agree, that one explanation, especially for possible plasmid encoded functions, may very well be transient carriage. To address this point we have added this line to the Discussion:

“It may be that the transient nature of AMR gene carriage is the result of rapid gain and loss during periods of differential selection, consistent with *S. flexneri* persisting in the environment where it is conceivable that selection by could be AMR less significant.”

*2) We consider the suggestion of a different life style or a different life-cycle between* S. flexneri *and other* Shigella *species as a very interesting hypothesis and very important to the field, if true. However, such an interpretation of the genomics data ideally requires further experimental support. As far as we know,* Shigella *has no environmental reservoir and human is its only host. You cite a paper reporting the isolation of* S. flexneri *from the environment (*[16]*); however, similar numbers of* S. dysenteriae *were also isolated from these environmental samples and therefore, do not support a different life style of* S. flexneri *than other* Shigella *species. Furthermore, different virulence genes (ipaBCD) from the virulence plasmid were missing in these so-called environmental isolates, but were present in clinical isolates, suggesting avirulent phenotype of these isolates.*

*It would therefore be prudent to provide more evidence for an environmental persistence of* S. flexneri*, for example from genome sequences of food or environmental isolates. Are these available in your collection?*

Our intention with the inclusion of Baker et al., 2002 was to point towards the fact that *S. flexneri* could be isolated from water unlinked to any outbreak. We realize that we should have made this clearer and have updated the text to reflect our intention when citing that reference. We also realise that we should have been clearer about previous evidence for the isolation of *S. flexneri* strains that are linked to outbreaks from water sources – there are a number of publications that have linked cases/outbreaks of *S. flexneri* disease to water sources, and we have updated the text to reflect this. These include cases in China (He et al., 2012), India (46) and Thailand (52). In all of these other cases, not only was *S. flexneri* isolated from the environment and cause of local disease; in these studies it was also the only *Shigella* species isolated from the water sources examined. Additionally, we have also referenced a further study conducted in Bangladesh (14) which demonstrated that a) *S. flexneri* risk areas are consistent based on geography (suggesting that the primary risk factors for *S. flexneri* disease in their study area are environmental, compared to *S. dysenteriae* where the risk factors were temporal and not linked to geographic location) and b) that there was a correlation between *S. flexneri* risk and the type of area that an individual lived in, which was directly related to likely exposure to water (flood controlled/not flood controlled).

With regard to our samples over 85% were isolated from human clinical cases for this study, while the remaining strains (which are all historical) are likely to be of clinical origin, although the metadata for these strains is often confusing and poorly recorded. Environmental isolates are not routinely collected especially in endemic regions and none were collected by our partner institutes over the time period covered by this study and in fact it is surprisingly hard to isolate many bacterial agents of diarrheal disease directly from the environment except in close proximity to latrines or where people practice open defecation. This is however very much in our future plans for prospective collection of true environmental isolates. However, collectively we believe that our data and previous published work provides a large body of data to support to the idea that a) *S. flexneri* can be isolated from the environment and b) that *S. flexneri* from an environmental source can be linked to clinical disease. To address these points we have added the references detailed above, with some additional text to our Discussion:

“It has previously been shown that *S. flexneri* strains can be recovered […] and has been doing so over the long term, in countries around the world.”

*Further, it might be useful to check for clues in the genome why* S. flexneri *might be more persistent in the environment. Are there any differences in carbon or nitrogen utilization pathways (different number of pseudogenes) in relation to* S. Sonnei*?*

*Alternatively, we suggest a test in the lab. A controlled comparison of the survival of* Shigella *species in water can provide strong experimental support for such an assumption and possibly explain the different epidemiology of* S. flexneri *vs.* S. Sonnei*.*

*In case that neither the bioinformatic analysis provide additional clues nor the experimental approach is feasible, we would ask you to clearly explain the caveats associated with your assumption of different life styles*.

We agree and one of the reasons serotyping is so useful/widely used for distinguishing *Shigella* and *E. coli* is because of the considerable metabolic overlap between members of the *E. coli* species complex. As mentioned by you (Reviewer 1), *Shigella* are essentially specialized subtypes of *E. coli*, and as such there are a very limited number of reactions that separate different *Shigella* species and S*higella* from *E. coli*. A more detailed genomic analysis (such as a pseudogene identification/comparison across the *Shigella* species, metabolic reconstruction) is warranted but would need to be performed in the context of all species and across *E. coli* too. This is a huge data set and considerable amount of work to do in a meaningful way.

In relation to the survival of *Shigella* in water we did not intend to suggest that the reason for the difference in prevalence was due to an inability of *S. sonnei* to survive at all in the environment. We have updated the text to make this distinction clearer. We agree that survival experiments would be fascinating but believe that in order to answer the question being posed would go considerably beyond the scope of this paper and would represent a significant amount of extra work to ensure that the experimental design for them was appropriate, and to provide results that are any more meaningful than data that already exists within the published literature. We had already cited work (Uyakik, Yazgi and Ayyildiz, 2008) that has examined the question of long term survival of *S. flexneri* in water, and found that in slightly saline water, S*. flexneri* could survive for >40 days in laboratory conditions. In our response to point 2 above we have added additional references around the established ability *S. flexneri* could survive and grow on various foodstuffs (25) along with further work (26, 22) that has similarly demonstrated that *S. flexneri* can survive in aqueous environments outside the host. Collectively, therefore, we believe that additional references and caveats will better enable the readership to assess the likely veracity of our hypothesis and to facilitate other groups who may wish to investigate the possibility of our idea for themselves.

*3) A slight weakness is that recent* S. flexneri *isolates from European origin are underrepresented in the collection. It appears that the most recent European strain (E2610/72) was from 1972. As global tourism from endemic countries to Europe increased greatly in the last 40 years or so, facilitating a possible intercontinental transmission might be overlooked if recent isolates are not included. For example of a recent outbreak in Europe, please see: M L Borg Ongoing outbreak of* Shigella flexneri *serotype 3a in men who have sex with men in England and Wales, data from 2009-2011. Eurosurveillance, Volume 17, 2012. Accordingly, we should like to ask you to assess the possibility to add more recent isolates from Europe in your study*.

We agree with the importance of thinking about recent European isolates, and in that respect we point towards our recent article focusing exclusively on the genomic epidemiology of the *Shigella flexnei* 3a outbreak mentioned by the reviewer (5), and referenced within our paper as part of our response to revision 1. The MSM paper details the clonal expansion of specific *Shigella flexneri* 3a clones linked to specific epidemiological, antimicrobial resistance and behavioral factors in MSM where *Shigella* is behaving non-traditionally as a sexually transmitted infection.

The current article is looking at a very different population, and in so doing provides much needed context for this type of study. However, while we agree with the importance of European isolates we also believe that it should be understood that the focus of the work presented here is not the spread of disease within the MSM community, but rather detailing the patterns of transmission within endemic/low income regions (where the vast majority of the global disease burden of this pathogen continues to be found). On this basis we believe that the addition of strains from Europe – which would almost entirely be associated either with travel or the MSM community – would not add anything to the paper, and, in fact, would replicate work that has already been done elsewhere.